# Comparison of the Life-Sustaining Treatment, Cardiopulmonary Resuscitation, and Palliative Care Implementation Rates between Homebound Patients with Malignant and Nonmalignant Disease Who Died in an Acute Hospital Setting: A Single-Center Retrospective Study

**DOI:** 10.3390/healthcare12020136

**Published:** 2024-01-08

**Authors:** Hisayuki Miura, Yuko Goto

**Affiliations:** Department of Home Care and Regional Liaison Promotion, National Center for Geriatrics and Gerontology, Obu 474-8511, Aichi, Japan; gotoyuko@ncgg.go.jp

**Keywords:** life-sustaining treatment, cardiopulmonary resuscitation, palliative care, in-hospital death, homebound patient

## Abstract

Objective: This study investigated and compared the implementation of life-support treatment (LST), cardiopulmonary resuscitation (CPR) implementation rates, and the influence of acute illnesses on the introduction of palliative care (PC) to homebound patients with malignant and nonmalignant disease, who subsequently died in an acute hospital setting. Methods: Among the homebound patients admitted to the ward in our hospital from 2011 to 2018, we investigated and compared the attributes, underlying diseases, causes of death, and rates of implementation of LST, CPR, and PC between patients with malignant and nonmalignant disease who died in the ward, using data obtained from hospitalization records. Furthermore, acute illnesses related to the introduction of PC were examined. Results: Of the 551 homebound patients admitted to the ward of an acute hospital, 119 died in the ward. Of the deceased patients, 60 had malignant disease and 59 had nonmalignant disease. Patients with nonmalignant disease had higher rates of LST implementation and CPR and a lower rate of PC. Patients with infectious disease, who required antimicrobial drugs, had significantly lower PC introduction rates. Conclusion: Understanding the influence of the timing of PC introduction in acute care for homebound patients with advanced chronic illness are issues to be considered.

## 1. Introduction

Japan has become a super-aging society, and the annual number of deaths in Japan is expected to exceed 1.6 million in 2040 [1]. The number of older individuals nearing end-of-life at home or in long-term care facilities who are receiving their preferred medical care is gradually increasing [2]. Medical needs have been changing significantly as the number of older patients with multiple chronic diseases increases. The concept of converting from the current medical system, which unevenly distributes acute care, to a system that enhances medical care for patients, so they can live in a manner similar to their previous daily lives, has been promoted [3]. Moreover, along with an increase in older patients, the number of patients with dementia and frailty has been increasing [4], and so has the number of patients with difficulty in expressing their pain. Under these circumstances, the Japanese government has been promoting home medical care (HMC) through multidisciplinary collaboration centered on physicians [5]. HMC is provided to more individuals who are in the advanced stages of chronic malignant and nonmalignant diseases than to individuals in the general older population. A multicenter prospective study in Japan reported that only one-third of patients receiving HMC died in their homes [6], regardless of their preferred places of death, although the factors associated with unexpected admissions in older homebound patients receiving HMC have been assessed in previous studies [7,8]. In Japan, policies have been promoted to encourage home care support hospitals and home care logistical support hospitals for homebound patients [9]. Because these are mostly general hospitals, only a few homebound patients can receive palliative care (PC) by experts. In Japan, PC for patients with cancer and acquired immunodeficiency syndrome (AIDS) is covered by medical insurance, and PC for malignancies is well-established. However, the development of PC methods for nonmalignant diseases has been delayed [10], and even opioid treatment methods for patients with nonmalignant disease, such as those with advanced respiratory disease, have not been established to date [11]. In particular, in acute-care settings for patients with advanced chronic illness, poor-quality PC by non-experts globally also has been reported [12,13]. The need for acute PC became globally recognized during the COVID-19 pandemic after 2020 [14,15], and PC guidance has been published by the academic society for PC in Japan [16].

Basically, all patients, regardless of whether they have chronic or acute nonmalignant or malignant disease, have equal rights to receive PC. The practice of PC is essential even in acute care hospitals, where many specialists for organ-specific diseases work. In Japanese HMC, life-sustaining treatment (LST), cardiopulmonary resuscitation (CPR), and PC are available upon a physician’s prescription, and surveys of current treatment states in patients who died at home have been performed previously [17,18]. However, no studies to date have reported the actual conditions of LST and PC in acute general hospital settings, in which many homebound patients with advanced chronic diseases have died.

Thus, this study investigated and compared the LST, CPR, and PC implementation rates as well as the influence of acute illnesses on introduction of PC to homebound patients with malignant and nonmalignant disease who subsequently died in an acute hospital setting.

This study aimed to clarify the current need to provide appropriate treatment and palliative care in acute hospital settings for terminally ill homebound patients in Japan, as well as the need for palliative care education for non-specialists.

## 2. Materials and Methods

### 2.1. Study Design and Population

This was a single-center retrospective study. The center (an acute hospital with approximately 300 beds) was one of the home care logistical support hospitals promoted by national policy and has a hospital ward named the “Home Medical Care Support Ward”, which only admits patients receiving regular physician-led home visits through neighboring clinics. In this ward, each physician who provided the relevant disease care for a patient during hospitalization performed the post-hospital treatment as the attending physician. The physicians, except for the PC team member, were not experts in PC. This study was conducted by implementing a mixed-method research design (convergent design) [19].

### 2.2. Data Collection

In this study, we enrolled patients aged ≥ 65 years who died after admission to this hospital ward between March 2011 and September 2018. We retrospectively collected data from the medical records during the last hospitalization of the deceased patients to obtain the following patient information: sex, age, origin organ in patients with malignant disease, underlying diseases in patients with nonmalignant disease, purpose of hospitalization at the last admission, cause of death, length of hospital stay, comorbidities, complications of infectious disease, activities of daily living (ADL), frailty, cognitive impairment, and nutritional status.

The Charlson comorbidity index [20], Barthel index [21], clinical frailty scale [22], and controlling nutritional status score [23], based on the information from medical records, were used to assess comorbidities, ADL, frailty, and nutritional state.

Cognitive impairment was defined as a dementia diagnostic factor, based on the medical records or long-term care insurance, with a level of “ADL of people with dementia” greater than III, which is used to assess cognitive impairment in long-term care insurance in Japan, as previously reported [24].

The following items were examined for the LST performed from hospitalization to death: oxygen use, peripheral infusion, artificial nutrition, antimicrobial drug use, noninvasive positive pressure (NPPV), mechanical ventilator, vasopressor, artificial dialysis, and blood transfusion. For CPR, the presence or absence of recorded discussion about the goals of care (GOCs) by the attending physicians and the actual implementation of CPR were checked. We explored the entries of GOC discussions across medical records during the hospitalization of all enrolled patients.

### 2.3. Definition of Terms

In this study, we defined a GOC discussion as an interactive discussion between a physician and patient/family involving either LST, CPR, or PC at the terminal stage. For PC, symptomatic management with analgesics and opioids, sedative drugs, and support from the PC team, including decision-making support, were checked. Patients who received PC were considered to have had any aforementioned treatments or interventions.

For patients who underwent CPR, their age group, underlying diseases, length of hospital stay (days), and cause of death were ascertained from the electronic medical records. A document analysis of the course leading to CPR was performed, according to the qualitative research method of Simons’ case study [25].

### 2.4. Ethical Considerations

This study complied with the principles outlined in the Declaration of Helsinki. This study was approved by the Institutional Review Board (IRB) of the National Center for Geriatrics and Gerontology (No. 675), according to the Japanese ethical guidelines for medical and health research involving human subjects [26]. Data were retrospectively obtained from medical records, and the need for written consent from patients and/or their families was waived by the IRB.

### 2.5. Statistical Analyses

Categorical variables are presented as numbers and percentages and the chi-square test was used to compare the two groups. Continuous variables are presented as means ± standard deviations or medians (interquartile ranges (IQRs)). The Shapiro–Wilk test was first applied to analyze whether normally distributed quantitative data could be analyzed using the unpaired t-test. Data that were not normally distributed were analyzed using the Mann–Whitney test to compare the two groups.

All statistical analyses in this study were performed using Statistical Package for the Social Sciences (version 29; IBM Corp., Armonk, NY, USA)

## 3. Results

### 3.1. Study Population

Of the 551 homebound patients admitted to the hospital ward, 119 (21.4%) died in the ward. Forty-five hospital physicians from 15 departments treated them. Of the deceased patients, 60 had malignant disease and 59 had nonmalignant disease.

Table 1 shows the origin organs for malignancy, underlying diseases for nonmalignant diseases, and causes of death. The gastrointestinal system was the most common origin organ in patients with malignant disease, and cerebrovascular diseases and dementia were the most common underlying diseases in patients with nonmalignant disease. The causes of death, except for malignancy itself, in patients with malignant disease were infection (e.g., inflammatory disease of the lungs), organ failure, cerebrovascular disease, and unknown. Meanwhile, the causes of death in patients with nonmalignant disease were organ failure, infection (e.g., inflammatory disease of the lungs), dementia/old age, neurological diseases, and others. Thus, besides exacerbation of the primary disease, complications from infectious diseases were the leading cause of death.

### 3.2. Purpose of Hospitalization at the Last Admission

Table 2 shows the purpose of hospitalization for patients with malignant and nonmalignant diseases.

In patients with malignant disease (*n* = 60), the purposes of hospitalization were exacerbation of the underlying disease (56.7%), complications of infectious diseases (16.7%), end-of-life care (13.3%), complication of other diseases (other than infection) (10.0%), and adjustments for home care (3.3%). In patients with nonmalignant disease (*n* = 59), the purposes of hospitalization included complications of infectious diseases (32.2%), respiratory failure (11.9%), heart failure (10.2), exacerbation of the underlying disease (other than respiratory and heart failure) (10.2%), and complications of other diseases, etc. (35.7%). In both groups, complications of infectious diseases (total *n* = 29, 24.3%) were ranked as the major reason for hospitalization.

### 3.3. Comparison of the Baseline Characteristics between Patients with Malignant and Nonmalignant Disease

Table 3 shows the characteristics of the patients with malignant and nonmalignant disease. The patients with nonmalignant disease were significantly older and had significantly longer lengths of hospital stay and lower Charlson comorbidity index and clinical frailty scale scores than those of the patients with malignant disease. Furthermore, the patients with nonmalignant disease also demonstrated a significantly higher rate of cognitive impairment.

### 3.4. Comparison of Practical Situations of Life-Sustaining Treatment, Cardiopulmonary Resuscitation, and Palliative Care between Patients with Malignant and Nonmalignant Disease

Table 4 shows the practical situations for the implementation of LST, CPR, and PC for patients with malignant and nonmalignant disease. For LST, the use of oxygen and peripheral infusion was >70% in both groups. Patients with nonmalignant disease showed higher rates of antimicrobial drug use than patients with malignant disease (62.7% vs. 21.7%; *p* < 0.001). The total implementation rate of LST (23.7%) (other than oxygen use, peripheral infusion, and antibacterial drug use) was higher in patients with nonmalignant disease than in patients with malignant disease (8.3%; *p* = 0.022).

For CPR, the rate of explicit discussions about the GOCs was 51.7% and 39.0% in the patients with malignant and nonmalignant disease, respectively. CPR was performed in five patients with nonmalignant disease only (Table 4).

The total introduction rate of PC was lower in patients with nonmalignant disease (22.0%) than in patients with malignant disease (73.3%; *p* < 0.001). The rate of opioid use was lower in patients with nonmalignant disease than in patients with malignant disease (13.6% vs. 63.3%; *p* < 0.001).

### 3.5. Cases of Cardiopulmonary Resuscitation

Table 5 shows a summary of the cases of CPR. All patients had nonmalignant disease. In patients with case numbers from one to four, CPR was performed for sudden changes after their condition had stabilized due to acute treatment after hospitalization. Meanwhile, patient no. five was rushed to the emergency room due to cardiopulmonary failure of an unknown cause. After hospitalization, the patient had cardiopulmonary arrest and died following CPR.

### 3.6. Relationship between the Introduction of Palliative Care, Complications of Infectious Disease, and Use of Antimicrobial Drugs

Table 6 shows the relationship between the introduction of PC, complications of infectious disease, and the use of antimicrobial drugs. In the patients with complications of infectious disease, the PC introduction rate was significantly low (*p* < 0.001). Most patients with complications of infectious disease received antimicrobial drugs.

## 4. Discussion

This retrospective study used medical record data of homebound patients admitted to a single-center (acute hospital). During the observation period, 119 people died in the ward; of these, 60 patients had malignant disease and 59 patients had nonmalignant disease. The proportions of deceased patients with malignant and nonmalignant disease were 60 of 158 (38.0%) and 59 of 393 (15.0%), respectively. Patients with malignant disease showed a higher mortality rate. For both patient types, complications of inflammatory disease, except exacerbation of the causative disease, were the main reason for hospitalization (Table 1 and Table 2).

Non-expert physicians, as the attending physicians, provided PC to treated homebound patients who had advanced chronic diseases until their patients neared death, although some patients had support from the PC team. In the comparison of the implementation rates of LST, CPR, and PC between patients with malignant and nonmalignant disease, patients with nonmalignant disease showed higher implementation rates of LST (other than oxygen use, peripheral infusion, and antimicrobial use) and CPR and a lower rate of PC. Complications of infection in hospitalized patients were relatively common, and the PC introduction rate was significantly lower in these patients.

### 4.1. Life-Sustaining Treatment

The implementation rates (>70%) of oxygen therapy and peripheral infusion were high in both groups and were much higher than the implementation rates of oxygen therapy (34%) and peripheral infusion (20%) in end-stage patients with malignant disease who received home PC by specialized home-care clinics in the study by Hashimoto et al. [18]. In our study, the rate of oxygen therapy usage in patients with malignant disease was also high (80%) (Table 4), but palliative oxygen for patients with advanced cancer is not necessarily recommended globally [24,27,28]. Uronis et al. [28] reported that oxygen was not superior to air for relieving breathlessness in a meta-analysis of randomized controlled trials on combined hypoxemic and non-hypoxemic patients. Additionally, in nonmalignant diseases, such as dyspnea caused by advanced respiratory disease, the Japanese guideline [29] states that oxygen therapy does not necessarily ameliorate dyspnea in patients without hypoxemia. Regarding artificial hydration at end-of-life, appropriate use guided by ethical considerations is required globally [30,31,32]. Regarding physician and nurse attitudes toward artificial hydration in terminally ill patients with malignant disease, Miyashita et al. [33] reported that more physicians (non-experts in PC) answered that artificial hydration alleviated the sensation of thirst, whereas more PC-unit physicians and nurses answered that withholding artificial hydration alleviated several physical symptoms. In our study, the attending physicians were non-experts in PC, which could have led to the higher usage rates of oxygen and artificial hydration (Table 4). In the previous reports about end-of-life preferences of the general public in Japan, approximately half to two-thirds of the population preferred antibiotics and fluid drip infusions [34] in cancer, cardiac failure, and dementia, and this may reflect the tendency of preferences of the general Japanese population. Compared with HMC, oxygen use and peripheral infusion for terminal patients in hospitals can be provided as a convenience of use and as “routine work” for patients in poor conditions. Given the limited evidence that oxygen and fluid replacement can provide pain relief for terminally ill patients worldwide [24,27,28,35], this practice needs to be reconsidered in Japan. Palliative care education is particularly necessary for physicians who do not specialize in palliative care, to ensure that patients do not receive therapy that harms instead of helps.

The reason that patients with nonmalignant disease showed higher implementation rates of LST (other than oxygen use, peripheral infusion, and antimicrobial drug use) is unclear; however, one factor for this phenomenon is that the stage of death could not be specified because predicting the life prognosis in nonmalignant diseases is difficult in general. Longer hospital stays for patients with nonmalignant disease can also make the implementation rate of LST higher.

### 4.2. Cardiopulmonary Resuscitation

In the medical records, the rate of GOC discussions between physicians and patients/families about LST and CPR or PC during hospitalization was 45.3% of all patients who died. Gott et al. [13] reported that, in PC provided by “generalists” in a hospital setting, there is scarce documented evidence of discussions with patients/families regarding end-of-life issues, and the topic of how to support the “generalist” PC providers was identified as an issue. In our study, CPR was performed only in cases of sudden changes from a stable state or in patients in a state close to cardiopulmonary arrest (Table 5). The use of CPR is basically unfamiliar in the advanced stages of chronic diseases. Even considering there was a possibility that the medical records were not recorded fully, it is possible that the presence or absence of CPR was decided by the attending physician in some patients. When a patient’s condition suddenly changes, the attending physician must make a decision, regardless of whether a GOC discussion has already taken place [36]. As all patients who received CPR in this study were terminally ill and in the advanced stages of a chronic disease, such as dementia or neurological disease, the possibility of resuscitation may have been low and CPR may not be indicated. Our findings also suggest that CPR may be routinely performed for conditions where there are no indication for resuscitation in Japan.

### 4.3. Palliative Care

In the study by Hashimoto et al. [18], opioids were used in 73% of end-stage patients with malignant disease by specialized home-care clinics, and this rate was similar to the rate of opioid use (63.3%) in hospitalized patients with malignant disease in this study. However, the introduction rate of PC for hospitalized patients with nonmalignant disease in our study was markedly lower (22.0%) than that for patients with malignant disease (73.3%) (Table 4). The lower PC introduction rate was especially obvious in patients with complications of infectious diseases or who used antimicrobial drugs (Table 6). This suggests that even if the underlying disease is an advanced chronic disease, when acute diseases, such as infections, were complicated, the focus should be only on the administration of antimicrobial drugs, and PC can be neglected. In Japan, PC has been traditionally limited to cancer and AIDS due to medical fees. Recently, end-stage heart failure has become a target of PC; however, the development of PC skills for nonmalignant diseases, such as respiratory diseases and dementia, has been delayed [10]. Our study suggested that the early introduction of PC for patients with complications of infectious diseases or cognitive decline can be very difficult, and these are situations in which doctors in acute-care hospitals fall behind in providing care for patients. However, recently, there have been attempts to spread PC to other nonmalignant diseases, such as the announcement of guidelines for PC for respiratory diseases, including in home care [29,37]. The effectiveness of opioids in respiratory failure has also been investigated [11,38]. Regarding PC for acute respiratory disease caused by coronavirus disease 2019, the enhancement of acute palliative care is required [12], and guidelines have been issued in developed Western countries [15,39,40], such as the United Kingdom and Japan [16]. In these guidelines, several measures, such as opioid administration, are specified [15,16,39,40] for acute PC. In any case, to ensure that homebound patients receive the PC they need, enhancing PC by use of “generalist” (non-experts in PC) for acute or subacute diseases, such as complications of infectious diseases, will be necessary in Japan. In this study, patients with nonmalignant disease did not have GOC discussions with medical professionals. In older patients, the complication of dementia might be problematic for such discussions. In the cases in which patients and health care professionals have previously discussed their wishes for care during advance care planning before cognitive decline occurs, it may be easier to introduce PC.

### 4.4. Issues Related to Treatment and Palliative Care in Acute Hospital Settings for Terminally Ill Homebound Patients in Japan

This study aimed to clarify the current need to provide appropriate treatment and palliative care in an acute hospital setting for terminally ill homebound patients in Japan, as well as the need for palliative care education for nonspecialists. Compared to findings from international institutions [24,27,28,35], our findings suggest that LST is performed more than necessary while necessary palliative care is not being provided—particularly for patients with nonmalignant diseases. Based on these findings, Japan needs a system to prevent life-prolonging treatment that has no benefit at the end of life from being performed on patients with nonmalignant diseases and to enhance palliative care. Achieving this aim will require enhanced guidelines for palliative care for patients with nonmalignant diseases and enhanced palliative care education for nonspecialists working in palliative care.

### 4.5. Limitations

The retrospective study has some limitations related to unextractable factors and insufficient adjustment of confounding factors. This study dealt with data in a ward designated to receiving homebound patients. Although physicians span a wide range of departments, including internal medicine and surgery, whether the implementation rates of LST, CPR, and PC in this study reflect the nationwide trend is unclear. Further confirmation is needed in a larger multicenter study.

In this survey, we did not obtain precise information on whether or not the enrolled homebound patients received LST or PC from the clinic physicians prior to admission. Therefore, we could not clarify the continuity of LST and PC between the home and hospitals.

## 5. Conclusions

Based on our analysis of the current situation in an acute hospital setting in Japan, this study found that LST is performed more than necessary, while necessary palliative care is not provided, for patients in the advanced stage of chronic diseases—particularly patients with nonmalignant diseases. Japan needs a system to prevent life-prolonging treatment that has no benefit at the end of life and to enhance palliative care, especially for patients with nonmalignant diseases, possibly by improving education for non-specialists working in palliative care.

## Figures and Tables

**Table 1 healthcare-12-00136-t001:** Origin organs in malignancy, underlying diseases in nonmalignant diseases, and causes of death.

Patients with malignant disease (*n* = 60)			
Origin organ	*n* (%)	Cause of death	*n* (%)
Gastrointestinal system	20 (33.9)	Malignancy itself	46 (76.7)
Kidney/urinary tract	5 (8.5)	Infection (including inflammatory disease of the lungs)	7 (11.7)
Respiratory system	4 (6.8)	Organ failure	3 (5.0)
Hematological system	3 (5.1)	Cerebrovascular disease	3 (5.0)
Mammary gland	2 (3.4)	Unknown	1 (1.7)
Gynecological system	2 (3.4)		
Head and neck	1 (1.7)		
Unknown	1 (1.7)		
Patients with nonmalignant disease (*n* = 59)			
Underlying disease	*n* (%)	Cause of death	*n* (%)
Cerebrovascular disease	13 (21.7)	Organ failure	21 (35.6)
Dementia	11 (18.3)	Infection (including inflammatory disease of the lungs)	20 (33.9)
Respiratory disease	8 (13.3)	Dementia/old age	6 (10.2)
Heart disease	8 (13.3)	Neurological disease	3 (5.1)
Bone and joint disease	8 (13.3)	Cerebrovascular disease	3 (5.1)
Neurological disease	7 (11.7)	Others ^†^	6 (10.2)
Gastrointestinal disease	2 (3.3)		
Diabetes mellitus	1 (1.7)		
Chronic renal failure	1 (1.7)		

^†^ Others include acute abdomen, upper gastrointestinal bleeding, diabetes mellitus, airway bleeding, arteriosclerosis obliterans, and cardiopulmonary arrest.

**Table 2 healthcare-12-00136-t002:** Purpose of hospitalization.

Patients with Malignant Disease (Total *n* = 60)	*n* (%)	Patients with Nonmalignant Disease (Total *n* = 59)	*n* (%)
Exacerbation of the underlying disease	34 (56.7)	Complications of infectious diseases	19 (32.2)
Complications of infectious diseases	10 (16.7)	Respiratory failure	7 (11.9)
End-of-life care	8 (13.3)	Heart failure	6 (10.2)
Complications of other diseases (other than infection) ^†^	6 (10.0)	Exacerbation of the underlying disease (other than respiratory and heart failure)	6 (10.2)
Adjustments for home care	2 (3.3)	Complication of	
		cerebrovascular disease	5 (8.5)
		gastrointestinal symptom	4 (6.8)
		dehydration	4 (6.8)
		bone and joint disease	2 (3.4)
		dysphasia	2 (3.4)
		arteriosclerosis obliterans	2 (3.4)
		Others *	2 (3.4)

^†^ Other diseases include cerebrovascular disease, gastrointestinal, respiratory, and cardiovascular symptoms, and dehydration. Others * include skin disease and adjustments for home care.

**Table 3 healthcare-12-00136-t003:** Characteristics of patients with malignant and nonmalignant disease.

Variables	Total (*n* = 119)	Malignancy (*n* = 60)	Nonmalignancy (*n* = 59)	*p*-Value
Age (years), mean (SD)	81.73 (8.49)	79.38 (6.98)	84.12 (9.25)	0.001
Female, *n* (%)	61 (51.2)	26 (43.3)	35 (59.3)	0.081
Length of hospital stays (days), median (IQR)	10.00 (18)	6.00 (11)	16.00 (22)	0.004
Charlson comorbidity index, median (IQR)	3.00 (2)	3.00 (4)	2.00 (2)	<0.001
Barthel index, median (IQR) (missing data = 15)	0.00 (15)	2.50 (24)	0.00 (5)	0.647
Clinical frailty scale, median (IQR) (missing data = 11)	8.00 (1)	9.00 (0)	8.00 (0)	<0.001
Cognitive impairment, *n* (%), (missing data = 12)	35 (32.7)	10 (18.2)	25 (48.1)	0.004
Controlling nutritional status, median (IQR)	2.00 (2)	2.00 (2)	2.00 (2)	0.657

**Table 4 healthcare-12-00136-t004:** Practical situations for the implementation of life-sustaining treatment, cardiopulmonary resuscitation, and palliative care in patients with malignant and nonmalignant disease.

Variables		Total (*n* = 119) *n* (%)	Malignancy (*n* = 60) *n* (%)	Non-Malignancy (*n* = 59) *n* (%)	*p*-Value
LST	Oxygen	93 (78.2)	48 (80.0)	46 (76.3)	0.623
	Peripheral infusion	91 (76.5)	42 (70.0)	49 (83.1)	0.093
	Artificial nutrition	13 (10.9)	4 (6.7)	9 (15.3)	0.133
	Antimicrobial drug	50 (42.0)	13 (21.7)	37 (62.7)	<0.001
	Mechanical ventilator	2 (1.7)	0 (0)	2 (3.4)	0.150
	Vasopressor	8 (6.7)	0 (0)	8 (13.6)	0.003
	Blood transfusion	3 (2.5)	1 (1.7)	2 (3.4)	0.549
	Total implementation of LST (other than oxygen use, peripheral infusion, and antimicrobial drug)	19 (8.4)	5 (8.3)	14 (23.7)	0.022
CPR	Recorded discussion about GOCs	54 (45.3)	31 (51.7)	23 (39.0)	0.165
	CPR	5 (4.2)	0 (0)	5 (8.5)	0.021
PC	Symptomatic management (analgesic)	11 (9.2)	9 (15.0)	2 (3.4)	0.029
	Symptomatic management (opioid)	46 (7.9)	38 (63.3)	8 (13.6)	<0.001
	Sedative drug	15 (12.6)	12 (20.0)	3 (5.1)	0.014
	Support by palliative care team	25 (21.0)	17 (28.3)	8 (13.6)	0.048
	Total introduction of PC	57 (47.9)	44 (73.3)	13 (22.0)	<0.001

LST, life-sustaining treatment; CPR, cardiopulmonary resuscitation; PC, palliative care; NPPV, noninvasive positive pressure ventilation; GOCs, goals of care; NA, not applicable.

**Table 5 healthcare-12-00136-t005:** Cases of cardiopulmonary resuscitation.

Case No.	Age Group	Underlying Disease	Length of Hospital Stay (Days)	Cause of Death	CPR Status
1	70s	Neurological disease	7	Airway bleeding	The patient was originally receiving gastrostomy feeding and mechanical ventilation with tracheostomy. The patient was urgently hospitalized due to hemoptysis. After admission, the patient was treated with hemostatic agents and antimicrobial drugs, and the patient’s condition became stable. However, after that, cardiopulmonary arrest due to sudden massive hemoptysis occurred. The patient died after CPR.
2	80s	Dementia	24	Aspiration pneumonia	The patient was hospitalized due to dehydration. After being hospitalized, the patient was diagnosed with aspiration pneumonia and treated with antimicrobial drugs. The patient’s condition was stable for a while; however, the patient stopped breathing suddenly. The patient died after CPR.
3	80s	Dementia	6	Recurrence of cerebral infarction	The patient was admitted to the hospital due to cerebral infarction. After hospitalization, the patient developed upper gastrointestinal bleeding and died, despite CPR.
4	80s	Diabetes mellitus	50	Acute heart failure	The patient was hospitalized due to pneumonia and heart failure. The patient improved with antimicrobial drugs, among others, and was undergoing rehabilitation to return home; however, the patient suddenly stopped breathing. The patient died after CPR.
5	80s	Multiple cerebral infarction	1	Unknown	A visiting nurse found that the patient had cyanosis during a visit to the patient’s home. The patient was rushed to the emergency unit and hospitalized; however, cardiopulmonary arrest occurred immediately. The patient died after CPR.

**Table 6 healthcare-12-00136-t006:** Relationship between introduction of palliative care, complications of infectious disease, and use of antimicrobial drugs.

	Introduction of Palliative Care		*p*-Value
	Presence	Absence	
Complications of infectious disease, *n* (%)	6 (20.7)	23 (79.3)	<0.001
No complications of infectious disease, *n* (%)	51 (56.7)	39 (43.3)	
Use of antimicrobial drug, *n* (%)	13 (26.0)	37 (74.0)	<0.001
No use of antimicrobial drug, *n* (%)	44 (63.8)	25 (36.2)	

## Data Availability

All data generated or analyzed during this study are included in this article, and additional data are available from the corresponding author upon reasonable request.

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
