# Peer review of "Comparison of the Life-Sustaining Treatment, Cardiopulmonary Resuscitation, and Palliative Care Implementation Rates between Homebound Patients with Malignant and Nonmalignant Disease Who Died in an Acute Hospital Setting: A Single-Center Retrospective Study"

_healthcare, 2024, doi:10.3390/healthcare12020136_

Round 1

Reviewer 1 Report

Comments and Suggestions for Authors

Dear authors, 

I have read through your paper and must congratulate you on your extensive work. The topic is increasingly important (and yet still seems underappreciated), methodology used is sound and the results are clearly presented. I do, however, have a few comments regarding introduction and discussion sections in relation to the aims and goals of your study.

In the Introduction you explain well what you did in the study but the actual aim of the study is unclear -  i.e. why (with what purpose) did you decide to do the comparison in the first place. I believe every study should have a goal (e.g. improving current state/availabiltiy of specific care plans to patients/education of  treating physicians in PC?). It would be helpful if you stated the aims of your study clearly.

Discussion lacks any more profound interpretation of the results that might be important for future patient care (in regard to aims of the study?).

Thank you for considering my comments and best regards.

Author Response

RESPONSES to Reviewer 1

1. COMMENT: I have read through your paper and must congratulate you on your extensive work. The topic is increasingly important (and yet still seems underappreciated), methodology used is sound and the results are clearly presented. I do, however, have a few comments regarding introduction and discussion sections in relation to the aims and goals of your study.

RESPONSE:

Thank you for your feedback on our work. We have revised the manuscript according to your comments.

2.COMMENT: In the Introduction you explain well what you did in the study but the actual aim of the study is unclear, i.e. why (with what purpose) did you decide to do the comparison in the first place. I believe every study should have a goal (e.g. improving current state/ availability of specific care plans to patients/education of treating physicians in PC?). It would be helpful if you stated the aims of your study clearly.

RESPONSE:

Thank you for your comment. We have added a sentence to clarify the aim of the study (lines 74–76)

3.COMMENT: Discussion lacks any more profound interpretation of the results that might be important for future patient care (in regard to aims of the study?).

RESPONSE:

Thank you for this suggestion. We have added some sentences about the interpretation of the results (lines 331–343).

Reviewer 2 Report

Comments and Suggestions for Authors

Thank you for this article, which shows that there is a difference in the palliative care of patients with malignant diseases compared to patients without malignant diseases. The authors present data from a Japanese hospital, referring to the deceased patients. I have some minor comments to improve the quality of your manuscript:

1. The title of the manuscript is very cumbersome and long. I would shorten it considerably. Furthermore, you write in the title (as well as in several places in the abstract and text, Lines: 14-15, 17-18, 133-134, 221 as well as table 1): nonmalignant and malignant patients. I believe that this is a poor translation and should definitely be changed. Since you yourself write in some places: patients with malignant and non-malignant diseases, you should use this form everywhere in the text, as it is correct. 

2. The abstract is also difficult to understand because there is no structure and a lot of information is mixed up. Could you please structure it (Objective, Methods, Result, Conclusion) and present the data more clearly? I only understood the content of the abstract after reading the entire manuscript. 

3. Line 57: COVID-19 was a pandemic, not an epidemic.

4. Section 2.4: It is unclear how deceased patients can consent to data analysis. Do patients generally sign a consent form for the use of their data on admission? This should be described in more detail. It also seems strange when you thank the patients for participating in the study at the end of the manuscript. This should be deleted under Acknowledgments. 

5. You describe that you looked at a total of 551 patients, and included 119 who died. It is  superfluous to write in line 221 that 60 and 59 are almost the same number, as everyone knows. It would actually be interesting to find out which underlying diseases the patients who did not die had. So what proportion of all patients do the 60 and 59 people make up? Can you provide more data on this?

6. In Table 4 you have also listed NPPV and artificial dialysis, although no patient received this treatment. The added value of this information is questionable and should be deleted. You should write n (%) once above the column, then you do not need to write it after each treatment. It is not clear why DNAR appears in the table legend without being written about in the table. 

7. You should position yourself more clearly when you write about giving oxygen and fluids to dying patients. There is no indication for both therapies, as patients do not benefit from them and are more likely to be harmed (especially by artificial fluids). It should not be dismissed as "routine work", but more education is needed so that patients do not receive a therapy that does not help but harms. You can also cite the following work here: doi 10.1177/0884533617724741

8. When you write about the CPR that took place, you should emphasize that there was actually no indication for resuscitation for any of the patients. All of them were seriously ill and should not have been resuscitated. It doesn't matter whether there have been discussions or not in advance, the indication is given by the doctor and there is no indication for resuscitation in patients with dementia or even gastric feeding and ventilation via a tracheostoma.  

9. The conclusions are poorly presented and should be revised. It is not true when you write that LST was performed in some patients (it was over 80%!). You should perhaps draw a conclusion from the data and outline what points need to be addressed further in the future and what investigations are needed to improve patient care.

Comments on the Quality of English Language

Some passages are very well written (introduction, methodology, results), while other sections are written in very poor English (abstract, discussion, conclusion). A revision should take place here.

Author Response

RESPONSES to Reviewer #2

COMMENT: Thank you for this article, which shows that there is a difference in the palliative care of patients with malignant diseases compared to patients without malignant diseases. The authors present data from a Japanese hospital, referring to the deceased patients. I have some minor comments to improve the quality of your manuscript:

RESPONSE:

Thank you for your feedback on our work. We have revised the manuscript accordingly.

COMMENT1. The title of the manuscript is very cumbersome and long. I would shorten it considerably. Furthermore, you write in the title (as well as in several places in the abstract and text, Lines: 14-15, 17-18, 133-134, 221 as well as table 1): nonmalignant and malignant patients. I believe that this is a poor translation and should definitely be changed. Since you yourself write in some places: patients with malignant and non-malignant diseases, you should use this form everywhere in the text, as it is correct. 

RESPONSE:

Thank you for this suggestion. We have shortened the title (lines 2–7) and corrected the descriptions of nonmalignant and malignant patients throughout the manuscript.

COMMENT2. The abstract is also difficult to understand because there is no structure and a lot of information is mixed up. Could you please structure it (Objective, Methods, Result, Conclusion) and present the data more clearly? I only understood the content of the abstract after reading the entire manuscript. 

RESPONSE:

Thank you for your comment. We have added the suggested headings and clarified the structure (lines 12–26).

COMMENT3. Line 57: COVID-19 was a pandemic, not an epidemic.

RESPONSE:

Thank you for this comment. We have corrected this term (line 59).

COMMENT4. Section 2.4: It is unclear how deceased patients can consent to data analysis. Do patients generally sign a consent form for the use of their data on admission? This should be described in more detail. It also seems strange when you thank the patients for participating in the study at the end of the manuscript. This should be deleted under Acknowledgments. 

RESPONSE:

We have added a description of the waiver of informed consent (lines 123–128). Data were retrospectively obtained from medical records, and the need for written consent from patients and/or their families was waived by the local Institutional Review Board. We have also deleted this sentence from the Acknowledgements (line 380).  

COMMENT5. You describe that you looked at a total of 551 patients, and included 119 who died. It is superfluous to write in line 221 that 60 and 59 are almost the same number, as everyone knows. It would actually be interesting to find out which underlying diseases the patients who did not die had. So what proportion of all patients do the 60 and 59 people make up? Can you provide more data on this?

RESPONSE:

Thank you for this suggestion. We have added the proportion of deceased patients with malignant and nonmalignant diseases (lines 230–233).

COMMENT6. In Table 4 you have also listed NPPV and artificial dialysis, although no patient received this treatment. The added value of this information is questionable and should be deleted. You should write n (%) once above the column, then you do not need to write it after each treatment. It is not clear why DNAR appears in the table legend without being written about in the table. 

RESPONSE:

Thank you for your comments. We have corrected Table 4 accordingly.

COMMENT7. You should position yourself more clearly when you write about giving oxygen and fluids to dying patients. There is no indication for both therapies, as patients do not benefit from them and are more likely to be harmed (especially by artificial fluids). It should not be dismissed as "routine work", but more education is needed so that patients do not receive a therapy that does not help but harms. You can also cite the following work here: doi 10.1177/0884533617724741

RESPONSE:

Thank you for this comment. We have revised the manuscript to clarify our position: there is no indication for the two therapies, as patients do not benefit from them and are more likely to be harmed. We have also cited the article that you recommended (lines 270–274).

COMMENT8. When you write about the CPR that took place, you should emphasize that there was actually no indication for resuscitation for any of the patients. All of them were seriously ill and should not have been resuscitated. It doesn't matter whether there have been discussions or not in advance, the indication is given by the doctor and there is no indication for resuscitation in patients with dementia or even gastric feeding and ventilation via a tracheostoma.  

RESPONSE:

Thank you for your comment. We have revised the manuscript to emphasize that there was no indication for all patients receiving CPR (lines 292–298).

COMMENT9. The conclusions are poorly presented and should be revised. It is not true when you write that LST was performed in some patients (it was over 80%!). You should perhaps draw a conclusion from the data and outline what points need to be addressed further in the future and what investigations are needed to improve patient care.

RESPONSE:

Thank you for this suggestion. We have added a description of the points that need to be addressed in the future and what investigations are needed to improve patient care (lines 357–363).